# Role of Microalgae in the Recovery of Nutrients from Pig Manure

**Ana Sánchez-Zurano** [1,*][iD]**, Martina Ciardi** [1]**, Tomás Lafarga** [1]**, José María Fernández-Sevilla** [1][iD]**, Ruperto Bermejo** [2] **and Emilio Molina-Grima** [1]

1 Department of Chemical Engineering, University of Almería, 04120 Almería, Spain; martina.ciardi@studio.unibo.it (M.C.); tomaslafarga@gmail.com (T.L.); jfernand@ual.es (J.M.F.-S.); emolina@ual.es (E.M.-G.)
2 Department of Physical and Analytical Chemistry, University of Jaén, 23700 Linares, Spain; rbermejo@ujael.es
* Correspondence: asz563@ual.es

**Abstract:** Animal production inevitably causes the emission of greenhouse gases and the generation of large amounts of slurry, both representing a serious environmental problem. Photosynthetic microorganisms such as microalgae and cyanobacteria have been proposed as alternative strategies to bioremediate agricultural waste while consuming carbon dioxide and producing valuable biomass. The current study assessed the potential of the microalga *Scenedesmus* sp. to remove nutrients from piggery wastewater (PWW) and the influence of the microalga on the microbial consortia. Maximum N-NH$_4^+$ consumption was $55.3 \pm 3.7$ mg·L$^{-1}$·day$^{-1}$ while P-PO$_4^{3-}$ removal rates were in the range 0.1–1.9 mg·L$^{-1}$·day$^{-1}$. N-NH$_4^+$ removal was partially caused by the action of nitrifying bacteria, which led to the production of N-NO$_3^-$. N-NO$_3^-$ production values where lower when microalgae were more active. This work demonstrated that the photosynthetic activity of microalgae allows us to increase nutrient removal rates from PWW and to reduce the coliform bacterial load of the effluent, minimising both their environmental impact and health risks. Microalgae assimilated part of the N-NH$_4^+$ present in the media to produce biomass and did not to convert it into N-NO$_3^-$ as in traditional processes.

**Keywords:** *Scenedesmus*; waste treatment; biotechnology; photosynthesis; respirometry; biomass

## 1. Introduction

Animal production will increase because of an increasing population, expected to reach 9–10 billion people by 2050 [1]. Meat production is one of the main causes of greenhouse gas emissions [2] and inevitably causes large amounts of slurry, which is a serious environmental concern [3]. Pig manure has been traditionally used as a fertiliser in rural areas. Currently, Spanish regulations limit the utilisation of pig manure as an organic fertilizer to up to 170 kgN·ha$^{-1}$·year$^{-1}$ (Directive 91/676/CEE) and this causes waste management problems in regions where agricultural lands are scarce and high amounts of manure are produced.

Microalgae-bacteria consortia have been proposed as a strategy to process wastewater and pig manure because of their ability to recycle organic matter and nutrients [4]. Indeed, microalgae are capable of consuming 25 tnN·ha$^{-1}$·year$^{-1}$ and 2.5 tnP·ha$^{-1}$·year$^{-1}$ and simultaneously produce up to 200 tn·year$^{-1}$ of valuable biomass, which could be further used to produce biofertilizers and biostimulants for agriculture [5]. An added advantage of microalgae is that they fix atmospheric carbon dioxide, one of the main problems associated with agriculture and food production. However, two important issues must be considered when microalgae are used for piggery wastewater (PWW) treatment: (i) high ammonium concentrations, such as those present in PWW, can lead to ammonia toxicity [6], and (ii) microalgae can affect the microbial community structure that appears naturally in PWW [7].

The latter is of key importance as the composition of the microalgae-bacteria consortia is key for an efficient nutrient removal.

During the day, microalgae consume inorganic carbon, nitrogen, and phosphorus (as well as other compounds) to produce biomass while simultaneously releasing oxygen. Oxygen produced by microalgae is used by heterotrophic bacteria to oxidise organic matter into inorganic compounds [8], producing carbon dioxide that is consumed by microalgal cells [9]. However, the reality of these interactions is far more complex, with different microalgal and bacterial populations taking place at the same time, including the aerobic growth of heterotrophic biomass, denitrification by the anoxic growth of heterotrophic biomass, and nitrification by the aerobic growth of nitrifying bacteria (AOB and NOB) [10]. Different interactions occur between microalgae and nitrifiers in terms of N-NH$_4^+$ availability. These interactions are not yet fully understood and contradictory results have been reported [11,12]. Thus, further studies are needed to identify how the utilisation of microalgae affects the bacterial community that appears naturally in PWW and, therefore, the efficiency of the integrated process.

For many years, respirometry has been considered as a rapid approach to assess metabolic activities in an economic and reliable way. Respirometry-based methods have been applied in convectional wastewater treatment to characterise heterotrophic and autotrophic biomass under different operational and environmental conditions [13–16]. This strategy has also been applied to quantify photosynthesis and respiration rates of cultures of phototrophic organisms such as microalgae and cyanobacteria [17–19]. More recently, techniques based on respirometry for activated wastewater treatment and phototrophic axenic cultures have been adapted to the microalgae-bacteria consortia that appear in wastewater [8,20,21].

The main goals of the current study were to provide a better understanding of the microalgae-bacteria interactions that occur in the microalgae-based PWW treatment processes and to assess the nutrient removal efficiency of the microalga *Scenedesmus* sp., widely studied because of its resistance to a wide range of environmental conditions.

## 2. Materials and Methods

### 2.1. Microalgae and Culture Conditions

*Scenedesmus* sp. has been widely studied for outdoor microalgae production and wastewater treatment. This strain was previously isolated from freshwater used in greenhouse fertigation by our research group and is, therefore, adapted to the local climate. The selected strain can grow well at pH, temperature, and salinity values ranging between 7–10, 26–40 °C, and 0–5 g NaCl·L$^{-1}$ [22]. Stock cultures were maintained photo-autotrophically in 1.0 L capacity photobioreactors using an Arnon medium [23]. Cultures were continuously bubbled with air—1.0% CO$_2$ mixture to control the pH at 8.0 ± 0.2. The culture temperature was kept constant at 22 ± 1 °C by regulating the air temperature in the chamber. The culture was artificially illuminated in a 12:12 h light:dark cycle using four Philips PL-32W/840/4p white-light lamps, providing an irradiance of 750 µE·m$^{-2}$·s$^{-1}$ on the photobioreactors surface. The average composition of the control medium and the piggery wastewater used is listed in Table 1.

### 2.2. Photobioreactors

Experiments were carried out in 1.0 L capacity lab-scale stirred-tank photobioreactors made with polymethylmethacrylate (0.08 m in diameter and 0.20 m height). To facilitate the up-scaling of the process, reactors were operated simulating outdoor raceway bioreactors. Two set of experiments were performed in triplicate (Figure 1). In the first set of experiments, photobioreactors were operated under either light or dark conditions and were fed 5-fold diluted PWW. Cultures produced in light or dark conditions were termed L-5 and D-5, respectively. The procedure was repeated but using 25-fold diluted PWW as the culture medium. In this case, cultures produced in light or dark conditions were termed L-25 and D-25, respectively. In both cases, the cultures were inoculated with *Scenedesmus*

sp. at an initial concentration of 0.5 g·L$^{-1}$ and were operated in batch mode for 6 days followed by operation in continuous mode by replacing daily 20% of the cultures volume with fresh PWW for 10 days, when the steady state was reached. Dissolved oxygen (DO) was controlled below 200%Sat by on demand air supply. The pH was controlled at 8.0 ± 0.2 by on-demand injection of $CO_2$.

**Table 1.** Average composition of the culture medium and piggery wastewater used as the influent in the bioreactors. Concentrations expressed as mg·L$^{-1}$.

| Parameters | Piggery Wastewater | Arnon |
|---|---|---|
| pH | 8.1 ± 0.3 | 7.5 ± 0.2 |
| COD | 2181.7 ± 100.9 | 16.0 ± 1.2 |
| Nitrogen-Nitrate | 56.4 ± 2.7 | 140.0 ± 4.5 |
| Chloride | 2060.2 ± 23.5 | 78.9 ± 2.1 |
| Potassium | 1800 ± 1.6 | 325.1 ± 6.3 |
| Calcium | 350.1 ± 0.2 | 364.9 ± 5.5 |
| Magnesium | 108.2 ± 14.1 | 12.2 ± 0.6 |
| Phosphorus-Phosphate | 119.2 ± 5.1 | 39.3 ± 3.1 |
| Nitrogen-Ammonium | 1485.6 ± 17.7 | 0.0 ± 0.1 |
| Iron | 4.8 ± 0.01 | 5.0 ± 0.3 |
| Copper | 1.1 ± 0.1 | 0.02 ± 0.00 |
| Manganese | 2.6 ± 0.0 | 0.5 ± 0.02 |
| Zinc | 20.1 ± 0.2 | 0.06 ± 0.01 |
| Boron | 5.3 ± 0.1 | 0.4 ± 0.0 |

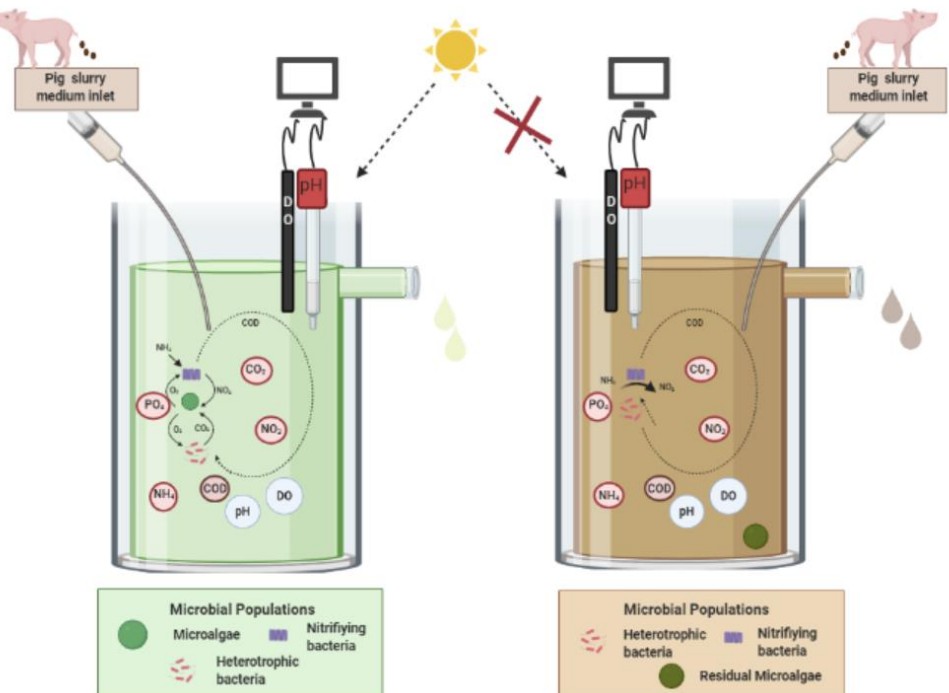

**Figure 1.** Graphical description of the experiments performed under light and dark conditions.

Photobioreactors were artificially illuminated using eight 28 W fluorescent tubes (Philips Daylight T5), programmed to mimic outdoor conditions: 12 h dark, 12 h light with a progressive increase in light intensity from 08:00 to 14:00 h. The maximum irradiance (PAR) inside the reactors in the absence of cells was 1000 µE·m$^{-2}$·s$^{-1}$, measured using an SQS-100 spherical quantum sensor (Walz GmbH, Effeltrich, Germany). Temperature was kept constant at 25.0 ± 1.0 °C.

### 2.3. Photosynthesis and Respiration

A photo-respirometer was used to obtain the microalgal net photosynthetic rate and the bacterial respiration rates in the photobioreactors under different operational conditions. The equipment consisted of an 80 mL jacketed transparent cylindrical glass flask, which was magnetically stirred and artificially illuminated using LED lamps. The photo-respirometer was also equipped with sensors for irradiance (QSL-1000, Walz, Germany), temperature (PT-100, Crison Instruments, Barcelona, Spain), pH (Crison 5343; Crison Instruments, Barcelona, Spain), and dissolved oxygen (Crison 5002; Crison Instruments, Barcelona, Spain), as well as a diffuser that allowed to control the flow rate of gases (air, $O_2$, $N_2$, and $CO_2$).

The protocol and methodology applied allowed us to distinguish between the metabolisms of the three main populations that appear in microalgae-bacteria wastewater: microalgae, heterotrophic bacteria, and nitrifying bacteria [8]. In the first place, microalgae-bacteria cultures were subjected to nutrient starvation (continuous light of 200 $\mu E \cdot m^{-2} \cdot s^{-1}$ and an aeration rate of 0.2 $v \cdot v^{-1} \cdot min^{-1}$) during 24 h to remove the organic matter and the ammonium present in the media. Then, culture samples were placed inside the photo-respirometer and subjected to four light–dark periods of 4 min each while the variation in DO under different conditions was measured and registered. During the light phases, photosynthetic microalgae generated oxygen, which was further consumed by endogenous respiration during darkness periods. The microalgae net photosynthesis rate was calculated as the difference between the slope of oxygen production during the light period minus the slope of oxygen consumption during the dark period. In the second place, culture samples were used to determine the heterotrophic respiration rate. For this purpose, 0.8 mL of sodium acetate (30.0 $g \cdot L^{-1}$) were added to the cultures before being subjected to four light–dark cycles of 4 min each. The respiration rate of the heterotrophic bacteria was calculated as the slope of oxygen consumption with sodium acetate minus the slope of the oxygen consumption during the dark period in the endogenous culture. Moreover, to determine nitrifying activity, 0.8 mL of ammonium chloride (3.0 $g \cdot L^{-1}$) were used as a substrate. As ammonium chloride can be consumed by both nitrifying bacteria and microalgae, two separate oxygen consumption rates were measured. The first one after addition of ammonium chloride alone, and the second one after addition of ammonium chloride and an allylthiourea solution (ATU), which was used as an ammonia-oxidizing bacteria inhibitor. ATU (1.0 $g \cdot L^{-1}$) was added until a concentration of 10 $mg \cdot L^{-1}$ and the nitrifying respiration rate was calculated as the difference between the total ammonium chloride respiration without ATU and the microalgae ammonium chloride respiration rate.

Finally, to correct the influence of oxygen desorption on the analytical determinations, the oxygen mass transfer coefficient was calculated using equation:

$$\frac{dC_{O2}}{dt} = K_L a \left( C_{O2}^* - C_{O2} \right), \tag{1}$$

where $\frac{dC_{O2}}{dt}$ is oxygen accumulation expressed as the derivate of $CO_2$ ($mg \cdot L^{-1}$) over time, $K_L a$ is the global oxygen mass transfer coefficient ($h^{-1}$), and $C_{O2}^*$ is the oxygen saturation concentration in the culture [8].

### 2.4. Bacterial Counts

Heterotrophic microbiota was calculated by plate count using Nutritive Agar in the steady state. An incubation time of 48 h at 30 °C was used to estimate the mesophilic aerobic microbiota [24]. Total coliforms and *Escherichia coli* in the steady state were quantified. Samples were diluted in phosphate buffered saline solution (PBS) to the decimal scale $10^{-4}$. Each dilution was inoculated in triplicate into sterile and disposable Petri dishes. Culture medium Cromocult® Coliform Agar (Merck KGaA, Gernsheim, Germany) was used. The Petri dishes were then incubated under controlled conditions at 36 °C for 24 h in the dark. Results were expressed as $CFU \cdot mL^{-1}$. The presence of *Salmonella* was evaluated by inoculating 10 mL of each sample into a flask with 50 mL Buffered Peptone Water (BPW) for pre-enrichment at 37 °C for 24 h. An aliquot of 0.1 mL was subsequently enriched in

10 mL of Rappaport Vassiliadis (RV) broth (Condalab, Madrid, Spain) at 42 °C during 48 h. Finally, to assess the presence of *Salmonella*-suspected colonies, each RV broth culture was plated onto Xylose Lysine Desoxycholate (XLD) (PanReac AppliChem, Barcelona, Spain) agar and incubated at 37 °C for 24 h.

*2.5. Statistical Analysis*

Results are the average of three independent experiments and are expressed as mean ± standard deviation (SD). Differences between samples were analysed using analysis of variance (ANOVA) with JMP 13 (SAS Institute Inc., Cary, NC, USA). A Tukey pairwise comparison of the means was conducted to identify where sample differences occurred. The criterion for statistical significance was $p < 0.05$.

## 3. Results

*3.1. Nutrient Removal*

Mass balances were conducted on the main nutrients (N-NH$_4^+$, N-NO$_3^-$, P-PO$_4^{3-}$, and COD) present in the reactors' inlets and outlets. The inlet concentration of N-NH$_4^+$ varied from 40–50 mg·L$^{-1}$ in L-25 and D-25 to 290–300 mg·L$^{-1}$ in L-5 and D-5, respectively ($p < 0.05$; Figure 2A). In the steady-state, the N-NH$_4^+$ concentrations in the outlet of the reactors were 3.4 ± 2.5, 3.6 ± 1.4, 94.6 ± 2.6 mg·L$^{-1}$, and 21.1 ± 1.4 in L-25, D-25, L-5, and D-5, respectively. N-NH$_4^+$ removal efficiency was significantly affected by both nutrient concentration ($p < 0.05$) and absence or presence of light ($p < 0.05$). The depuration efficiency of the N-NH$_4^+$ present in the most diluted culture media, L-25 and D-25, was greater than 92%. The cultures' N-NH$_4^+$ consumption was 8.5 ± 0.5 and 8.4 ± 0.3 mg·L$^{-1}$·day$^{-1}$ in L-25 and D-25, respectively. These values were lower than those obtained for L-5 and D-5, which were 40.5 ± 1.1 and 55.3 ± 3.7 mg·L$^{-1}$·day$^{-1}$, respectively ($p < 0.05$). The highest N-NH$_4^+$ removal was obtained in D-5 ($p < 0.05$).

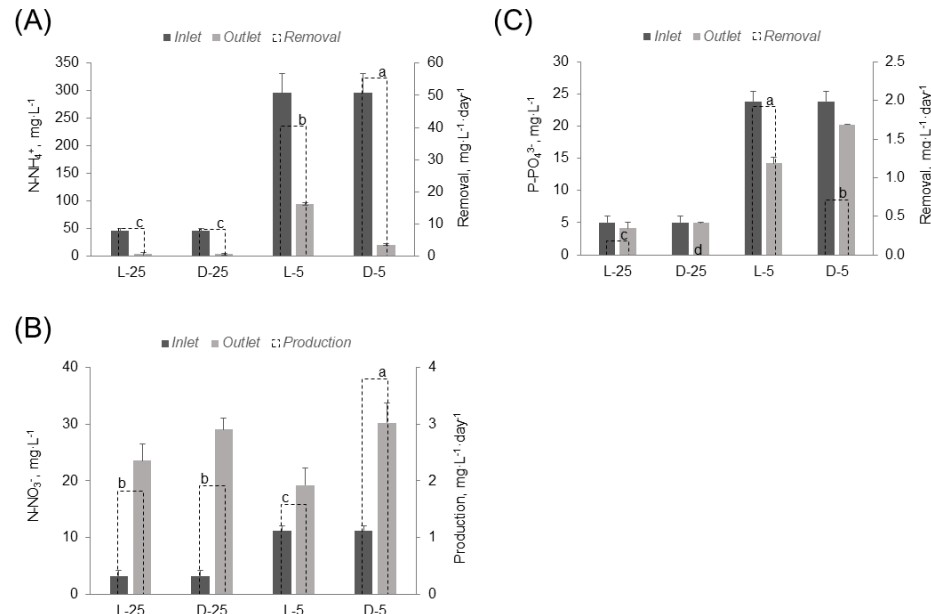

**Figure 2.** Inlet and outlet concentration and removal/production of (**A**) N-NH$_4^+$, (**B**) N-NO$_3^-$, and (**C**) P-PO$_4^{3-}$ in L-25, D-25, L-5, and D-5. Different letters indicate significant differences.

The second main nitrogen form in PWW was N-NO$_3^-$ (Figure 2B). The inlet concentration of N-NO$_3^-$ in the reactors varied from 3.3 mg·L$^{-1}$ in L-25 and D-25 to 11.3 mg·L$^{-1}$ in L-5 and D-5, respectively ($p < 0.05$; Figure 2B). The concentration of N-NO$_3^-$ was higher in the outlet than in the inlet ($p < 0.05$). N-NO$_3^-$ concentration in the outlet of the photobioreactors was 19.2 ± 0.5, 30.2 ± 2.5, 23.5 ± 3.0, and 29.1 ± 2.0 mg·L$^{-1}$ in L-5, D-5, L-25,

and D-25, respectively. These values represent a N-NO$_3^-$ production of 1.6 ± 0.1, 3.8 ± 0.5, 1.8 ± 1.3, and 1.9 ± 0.5 mg·L$^{-1}$·day$^{-1}$, respectively. N-NO$_3^-$ production was especially higher in D-5 ($p < 0.05$).

The current study also determined P-PO$_4^{3-}$ in the inlet and outlet of the reactors. Results are shown in Figure 2C. Significant difference in the inlets were observed, being 23.8 mg·L$^{-1}$ in L-5 and D-5, and 5.0 mg·L$^{-1}$ in L-25 and D-25, respectively ($p < 0.05$). The P-PO$_4^{3-}$ removal rate was calculated as 1.9 ± 0.1, 0.7 ± 0.2, 0.2 ± 0.1, and 0.1 mg·L$^{-1}$·day$^{-1}$ in L-5, D-5, L-25, and D-25, respectively. P-PO$_4^{3-}$ concentrations in the outlets were 14.2 ± 0.4, 20.3 ± 0.9, 4.1 ± 0.9, and 5.0 ± 0.1 mg·L$^{-1}$, respectively. In addition, P-PO$_4^{3-}$ removal rates corresponded to consumption efficiencies of 40 and 15% for L-5 and D-5 and of 18 and 0% for L25 and D-25, respectively.

Finally, the COD concentration of the reactors was also assessed (Figure 3). L-5 and D-5 reactors were fed with 436.3 mg·L$^{-1}$ while a significantly lower concentration was fed to L-25 and D-25 reactors, measured as 83.2 mg·L$^{-1}$ ($p < 0.05$). COD values in the outlets where 352.5 ± 14.8, 487.3 ± 0.2, 142.1 ± 5.6, and 133.5 ± 13.4 mg·L$^{-1}$ in L-5, D-5, L-25, and D-25. COD consumption was 16.7 ± 3.1 mg·L$^{-1}$·day$^{-1}$ for L-5 and no COD removal was observed in D-5, L-25, and D-25. Indeed, for these reactors, the outlet COD concentration was higher than in the inlet ($p < 0.05$).

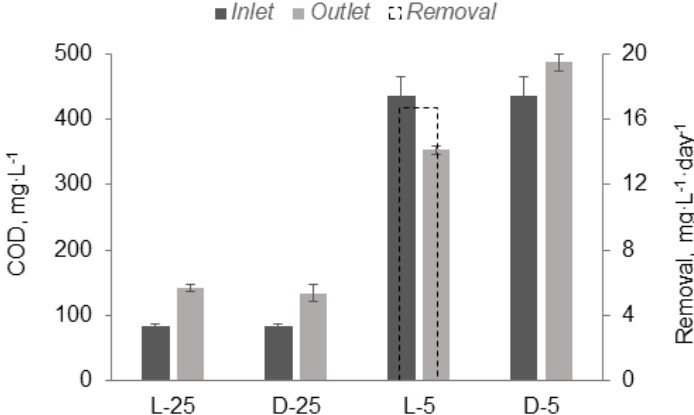

**Figure 3.** Inlet and outlet concentration and removal/production of COD in L-25, D-25, L-5, and D-5.

*3.2. Respirometric Analysis*

The net photosynthetic rate was 15.3 ± 0.7, 1.1 ± 0.5, 6.7 ± 0.8, and 0.3 ± 0.2 mg·L$^{-1}$·h$^{-1}$ in L-5, D-5, L-25, and D-25, respectively. Net photosynthesis was significantly affected by both nutrient concentration ($p < 0.05$) and absence or presence of light ($p < 0.05$). Both reactors operating under light conditions showed a higher photosynthetic rate, being higher in L-5 than in L-25, despite of a similar biomass concentration (Figure 4A). The heterotrophic bacteria respiration rate was 1.35 ± 0.11, 1.54 ± 0.21, 0.26 ± 0.12, and 0.33 ± 0.13 mg·L$^{-1}$·h$^{-1}$ in L-5, D-5, L-25, and D-25 (Figure 4B). Heterotrophic activity in L-25 was 5-fold lower than in L-5 ($p < 0.05$). The respiration rate of nitrifying bacteria was 1.4 ± 0.2, 2.5 ± 0.1, 0.5 ± 0.1, and 0.4 ± 0.1 mg·L$^{-1}$·h$^{-1}$ in L-5, D-5, L-25, and D-25. Higher oxygen consumptions were observed for samples diluted 5-fold when compared to 25-fold, being higher in D-5 than in L-5 ($p < 0.05$).

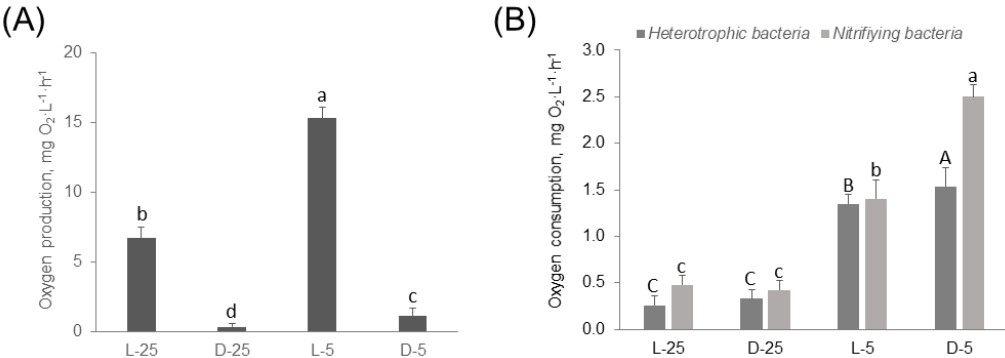

**Figure 4.** (**A**) Microalgae activity at different influent concentration and under light/dark conditions. (**B**) Heterotrophic and nitrifying activity at the different experiments. Different letters indicate significant differences.

### 3.3. Microbiologic Analysis

Heterotrophic bacterial counts were $2.35 \times 10^5$, $1.35 \times 10^5$, $2.5 \times 10^5$, and $1.75 \times 10^4$ CFU·mL$^{-1}$ in L-5, D-5, L-25, and D-25, respectively (Table 2). Coliforms were $7.8 \times 10^1$ and $4.11 \times 10^2$ CFU·mL$^{-1}$ in L-5 and D-5, respectively, and $2.6 \times 10^1$ and $2.05 \times 10^2$ CFU·mL$^{-1}$ in L-25 and D-25. Moreover, *E. coli* and *Salmonella* sp. were not detected (ND) in any sample.

**Table 2.** Microbial population counts during the experiments. Data are expressed as CFU·L$^{-1}$.

|  | L-25 | D-25 | L-5 | D-5 |
| --- | --- | --- | --- | --- |
| Heterotrophic bacteria | $2.50 \times 10^5$ | $1.75 \times 10^4$ | $2.35 \times 10^5$ | $1.35 \times 10^5$ |
| Coliforms bacteria | $2.60 \times 10^1$ | $2.05 \times 10^2$ | $7.80 \times 10^1$ | $4.11 \times 10^2$ |
| *E. coli* | ND | ND | ND | ND |
| *Salmonella* sp. | ND | ND | ND | ND |

## 4. Discussion

Nutrient removal from waste streams using microalgae-bacteria consortia has been widely studied during the last couple of decades. This approach has been proposed as the key strategy to reduce microalgal biomass production costs to under 1–2 €·kg$^{-1}$ [24]. Investigations on microalgae-based bioremediation led to the understanding that nutrient removal is caused by assimilation, anaerobic ammonia oxidation, nitrification, and denitrification, among other processes. However, little is known about the specific contribution of microalgae to the process and their effect on the systems performance [25]. The current study aimed at understanding the influence of photosynthetic activity on nutrient consumption during PWW treatment. Results, shown in Figure 1, demonstrated that the microalgae-bacteria consortia allowed us to achieve high N-NH$_4^+$ removal rates. The major removal rates were observed in samples L-5 and D-5, attributed to higher N-NH$_4^+$ content in the inlet. Almost a complete ammonia removal was observed in L-25 and D-25. However, part of nitrogen in the form of N-NH$_4^+$ was converted to N-NO$_3^-$ by the action of nitrifying bacteria, obtaining higher content of N-NO$_3^-$ in the outlets than in the inlets. In the reactors operated under light conditions, the assimilation of N-NH$_4^+$ is caused by both microalgae and nitrifying bacteria. Microalgae use N-NH$_4^+$ to produce biomass while nitrifying bacteria use it to growth and to carry out the first step of nitrification. In the current study, the nitrifying activity predominated in the reactor operating under dark conditions since the phototrophic activity is negligible. As a result, the content of N-NO$_3^-$ in the outlet of reactors under light conditions is lower when compared to the systems that were maintained in dark, demonstrating a lower N-NO$_3^-$ production. These findings can be attributed to two main factors: (i) microalgal growth reduces AOB populations, and (ii) microalgae are capable of assimilating the N-NO$_3^-$ produced during nitrification. The latter is less probable because previous reports suggested that when N-NH$_4^+$ and

N-NO$_3$$^-$ are both present in the media, microalgae generally prefer the former [26,27]. PWW treatment processes allow adequate N-NH$_4$$^+$ removal rates but lead to an increase in the production of N-NO$_3$$^-$ and, therefore, to a loss of nutrients. The use of microalgae in wastewater treatment processes could be very favourable as these nutrients could be used for microalgal biomass production. Operating under light conditions, when microalgal phototrophic activity is enhanced, allowed us to avoid high nutrient losses and to obtain higher nutrient removal rates [10].

The phosphorus removal rates reported show that microalgal phototrophic activity (L-25 and L-5) increased phosphorous consumption from PWW. These values were lower when compared to those reported in a previous study during the treatment of 10- and 20-fold diluted PWW under indoor and outdoor conditions (81–99%) [4]. However, in that study, the authors operated with an hydraulic retention time (26 days) higher than the one assessed in the current study (5 days), and it is accepted that operational conditions have a significant impact on biomass productivity and nutrient removal rates, especially process duration [10]. In the current study, phosphorous consumption in L-5 was almost twice the value of L-25. To explain this difference, it is important to highlight that evaluating phosphorous uptake in microalgae-bacteria based systems is particularly difficult. Phosphorus removal is influenced by multiple environmental factors such as temperature or photoperiod [28]. Indeed, higher phosphorus removal rates were reported in summer than in winter [29]. Moreover, luxury phosphorous uptake phenomena has been reported at high phosphate concentrations in a mixed microalgal consortium dominated by *Scenedesmus* [30]. In this case, when phosphate aqueous concentration increased from 5 to 15 mg·L$^{-1}$, the microalgal acid soluble polyphosphate content increased up to three times [30]. In the experiments presented in this work, the environmental conditions such as temperature and light were kept constant. Thus, this difference in phosphorus consumption could be attributed to the phenomenon of luxury uptake since the biomass concentration reached by L-25 and L-5 was similar (around 0.6 g·L$^{-1}$). Therefore, phosphorus removal in microalgae-bacteria consortia involve phenomena including the assimilation by both microalgae and bacteria to form biomass and intracellular polyphosphate compounds and also phosphorous precipitation at high pH values (if it is not controlled) [31]. Phosphorous assimilation into algal-bacterial biomass was likely the main removal mechanism based on the adequate controlled pH values prevailing in the photobioreactors (pH = 8.0), which avoided phosphate precipitation [32].

The COD removal obtained in the experiments was particularly low. COD removal was only observed in L-5, allowing a removal rate of 20%. In this context, the fraction of readily biodegradable organic carbon in PWW influenced the COD removal, and difficulted the inter-studies comparison. Moreover, the biodegradability range from 0% to 80% in PWW due to farm swine manure management practices such as shed cleansing or waste storage conditions [33]. In the current study, the PWW used was kept in rafts for over a year, and therefore, most of the organic matter present could be not readily biodegradable.

A respirometric methodology was used to assess the main microbial metabolisms that appeared in microalgae-bacteria cultures under different PWW concentrations and light/dark conditions: microalgae, heterotrophic bacteria, and nitrifying bacteria. Results showed that microalgae activity under dark conditions was especially low, resulting in a minimal photosynthetic activity due to the residual microalgal cells in the photobioreactors. Although a higher activity was expected in L-25 than in L-5, since ammonium concentrations above 100 mgN·L$^{-1}$ have been reported as inhibitory for microalgae cultures [34], results showed the opposite effect. The net photosynthesis in the reactors under light conditions differed significantly between 5 and 25 times diluted PWW. The greater value was observed for L-5, despite a similar biomass concentration being achieved in both assays. The observed decrease in photosynthetic activity could have been caused by a limitation of micronutrients, which were present in very low concentrations in L-25 and D-25. Previous authors described that micro-elements (such as iron and manganese) have an important role on the growth and photosynthetic electron transport of microalgae [35,36]. Iron is

an essential element for photosynthesis and respiration in microalgae, whose growth is often limited due to the poor iron solubility [37]. In natural environments, many heterotrophic bacteria produce siderophores, small organic molecules that tightly bind to iron and thereby increase its solubility. Therefore, heterotrophic bacteria can solubilize iron, which could be available for microalgae because, to date, microalgae were not reported as siderophore producers [38]. Thus, the low photosynthetic activity in L-25 could have been caused by a low heterotrophic activity. Heterotrophic activity in L-25 was five times lower than the heterotrophic respiration measured in L-5. In turn, the low heterotrophic activity detected in L-25 could be explained by the limited biodegradable organic matter measured in the samples. Related to the respiration rate of nitrifying bacteria measured by respirometric techniques, results show that rates under light and dark conditions did not differ significantly when PWW diluted 25 times was used, which is in line with previous reports [39]. However, nitrifying activity varied between light and dark conditions when the stirred-tank reactors were fed with PWW diluted 5 times. This variability may be the result of the high microalgae activity measured in L-5, which could compete for the ammonium present in the medium with ammonium oxidizing bacteria [11].

Heterotrophic bacteria include all bacteria that use organic nutrients for growth. These bacteria are natural inhabitants of food, air, animal/human body, and all types of water. Within this group, both bacterial pathogens and coliforms (*Escherichia*, *Klebsiella*, *Enterobacter*, *Citrobacter*, and *Serratia*) are included [40]. Heterotrophic plate count (HPC) can be used for detection of all bacteria that consume organic compounds, but cannot be used as indicators of pathogenic conditions. In the samples, heterotrophic bacteria, coliforms bacteria, *E. coli*, and *Salmonella* sp. were measured in the outlets, after removing the microalgae-bacteria biomass. Results suggested that the microalgae-bacteria cultures under light conditions, when microalgae phototrophic activity was enhanced, presented a greater number of heterotrophic bacteria. This difference can be due to different factors. On the one hand, the use of microalgae in wastewater treatment involves many associations with other microorganisms present in wastewater. These associations have been described in the phycosphere, the microscale area surrounding microalgae cells where metabolites are exchanged between microalgae and bacteria [41]. The phycosphere is equivalent to an "oasis" for heterotrophic bacteria, where high concentrations of fixed organic carbon is excreted for consumption [42]. On the other hand, the stirred-tank reactors operated under light conditions achieved values of dissolved oxygen up to 200%, which can be consumed by heterotrophic bacteria. Therefore, increasing of phototrophic activity could have led to an increase in heterotrophic bacteria, because they form consortia that favour nutrient removal and biomass production. Moreover, results suggested that microalgae activity allowed for the reduction of the content of coliform bacteria as lower coliform bacteria were found in the reactors operated under light conditions. This was in line with previous publications that described that the environmental factors that are favourable for algal growth are unfavourable for the survival of coliforms [43].

## 5. Conclusions

This work demonstrated that the photosynthetic activity of microalgae allows us to improve the nutrient removal rates in PWW and to reduce the coliform bacterial load of the effluents. This was mainly caused by microalgae, which allowed $N-NH_4^+$ assimilation instead of converting it into $N-NO_3^-$, which occurs in traditional PWW due to the oxidizing ammonium activity. Microalgae utilisation also led to a reduction of the phosphorus present in the PWW due to its assimilation into microalgal biomass. The microalgae-bacteria consortia enhanced both the activity of microalgae that mainly consumed the N and P present in the PWW and the activity of heterotrophic bacteria that consumed organic matter. Further studies will include the up-scaling of the process outdoors and a complete characterisation of the microorganisms present in the consortia using metagenomic analyses.

**Author Contributions:** Conceptualization, T.L.; methodology, A.S.-Z. and M.C.; formal analysis, T.L. and A.S.-Z.; investigation, A.S.-Z. and M.C.; writing—original draft preparation, A.S.-Z.; writing—review and editing, T.L. and A.S.-Z.; visualization, T.L. and R.B.; supervision, J.M.F.-S. and E.M.-G.; funding acquisition, R.B. and J.M.F.-S. All authors have read and agreed to the published version of the manuscript.

**Funding:** This research was funded by the SABANA project (grant # 727874) of the European Union's Horizon 2020 Research and Innovation Programme and by the PURASOL project, funded by the Spanish Ministry of Economy and Competitiveness (CTQ2017-84006-C3-3-R). Authors also thank IFAPA, the Spanish Ministry of Education (FPU16/05996), the Spanish Ministry of Science, Innovation, and Universities (IJC2018-035287-I), and the BBVA Foundation.

**Institutional Review Board Statement:** Not applicable.

**Informed Consent Statement:** Not applicable.

**Data Availability Statement:** The data presented in this study are available on request from the corresponding author. The data are not publicly available due to privacy.

**Conflicts of Interest:** The authors declare no conflict of interest.

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
