# Peer review of "Role of Microalgae in the Recovery of Nutrients from Pig Manure"

_processes, doi:10.3390/pr9020203_

Round 1

Reviewer 1 Report

Dear Authors,   
 I think the manuscript is very interesting. The role of microalgae and bacteria in native/artificial consortia in wastewater is always a complex topic necessary to study to understand what happens in wastewater in terms o f biomass production and nutrient removal.
In my opinion, the manuscript is ready for publication after minore revisions. In detail, in par. 3.1 (lines 176-185) data are reported in confusing way: the concentration for N-NH4+ for L-25 and L-5 is the same in the text, but different in the figure (L-25 is higher than L-5). I suppose that because of this error, also data about nutrient removal efficiency should be correct. In line 190, data are referred to figure 2B ( in the brackets) and not 1B.

Author Response

Thank you very much for the comments and suggestions. We agree with you about the minor changes. Section 3.1 has been revised and the errors have been checked. Also, the error related with figure's numeration has been corrrected.

Reviewer 2 Report

Journal: MS ID: processes-1076698

The work demonstrates the photosynthetic activity of microalgae to increase nutrients removal rates in pig wastewater as well as reduction the coliform bacteria population in the effluent minimising both, their environmental impact and health risks. Microalgae culture assimilated part of the N-NH4+ present in the media to produce biomass and did not to convert it into N-NO3-, as occurs in traditional processes via ammonium oxidizing activity. These results are due to the microalgae-bacteria consortia that enhanced both the activity of microalgae that mainly consumed the N and P present in the pig wastewater, and the activity of heterotrophic bacteria that consumed organic matter. As concerns methodology I really appreciate the use of this relatively novel and useful technique to distinguish microalgae net photosynthesis rate and bacteria respiration rates in co-cultures present in photobioreactors.

This MS is well prepared, the objectives are topical, and presented mostly logically, straightforward. The MS is of good quality, based on well-designed experiments with number of data. I do not have any major objection against the text. Some of them to be considered I have suggested directly in the MS text (see enclosed file). From my point of view these might improve the text and its better understanding. Some parts of the MS are yellow highlighted as they may need rephrasing.

Here I list the most pertinent:

  • First of all, in some parts of the MS English style is not perfect. It needs careful reading and correction by native speaker (or proficient person).
  • The Scenedesmus strain should be better described/identified.
  • Character size in all figures should be enlarged/increased to be better legible; the form mg L-1, mg (O2) L-1 h-1 etc. should be used as in the MS text.
  • In section 2.6 Statistical analysis is mentioned, but I have not found any statistics in figures shown by characters of significance. It might be good to add if it was performed.

Author Response

Thank you very much for the comments and suggestions. We agree with you about the minor changes:

1. First of all, in some parts of the MS English style is not perfect. It needs careful reading and correction by native speaker (or proficient person).
Response: The english style has been checked in order to improve the quatlity of the manuscript. The changes are remarked in red color.
2. The Scenedesmus strain should be better described/identified.
Response: This point has been improved and completed in materials and methods.
3.Character size in all figures should be enlarged/increased to be better legible; the form mg L-1, mg (O2) L-1 h-1 etc. should be used as in the MS text.
Response: the character size of the figures have been enlarged.
4. In section 2.6 Statistical analysis is mentioned, but I have not found any statistics in figures shown by characters of significance. It might be good to add if it was performed.
Response: The statistical analysis has been specified in the figures.